# The Molecular and Genetic Mechanisms of Inherited Bone Marrow Failure Syndromes: The Role of Inflammatory Cytokines in Their Pathogenesis

**DOI:** 10.3390/biom13081249

**Published:** 2023-08-16

**Authors:** Nozomu Kawashima, Valentino Bezzerri, Seth J. Corey

**Affiliations:** 1Departments of Pediatrics and Cancer Biology, Cleveland Clinic, Cleveland, OH 44195, USA; kawashn@ccf.org; 2Cystic Fibrosis Center, Azienda Ospedaliera Universitaria Integrata, 37126 Verona, Italy; valentino.bezzerri@univr.it

**Keywords:** Fanconi anemia, Shwachman–Diamond syndrome, dyskeratosis congenita, inflammatory cytokines

## Abstract

Inherited bone marrow failure syndromes (IBMFSs) include Fanconi anemia, Diamond–Blackfan anemia, Shwachman–Diamond syndrome, dyskeratosis congenita, severe congenital neutropenia, and other rare entities such as GATA2 deficiency and SAMD9/9L mutations. The IBMFS monogenic disorders were first recognized by their phenotype. Exome sequencing has validated their classification, with clusters of gene mutations affecting DNA damage response (Fanconi anemia), ribosome structure (Diamond–Blackfan anemia), ribosome assembly (Shwachman–Diamond syndrome), or telomere maintenance/stability (dyskeratosis congenita). The pathogenetic mechanisms of IBMFSs remain to be characterized fully, but an overarching hypothesis states that different stresses elicit TP53-dependent growth arrest and apoptosis of hematopoietic stem, progenitor, and precursor cells. Here, we review the IBMFSs and propose a role for pro-inflammatory cytokines, such as TGF-β, IL-1β, and IFN-α, in mediating the cytopenias. We suggest a pathogenic role for cytokines in the transformation to myeloid neoplasia and hypothesize a role for anti-inflammatory therapies.

## 1. Introduction

Bone marrow failure refers to the inadequate production of healthy blood cells and presents throughout the age spectrum. Patients with bone marrow failure present with mono-, bi-, or trilineage cytopenia(s) in the peripheral blood, and their bone marrow shows hypoplastic, normal, or hyperplastic cellularity with reduced function of producing mature blood cells. Transient bone marrow failure may be triggered by exposure to chemical agents, radiation, or viral infection. Chronic bone marrow failure, lasting more than three months, may be classified into two major categories: acquired aplastic anemia and inherited bone marrow failure syndromes (IBMFSs). Either disorder needs to be distinguished from hypoplastic myelodysplastic syndrome (MDS). A rare condition, aplastic anemia results from the post-natal acquisition of autoimmune-mediated (especially, T cell immune-mediated) destruction of hematopoietic stem and progenitor cells (HSCs/HSPCs). One condition associated with aplastic anemia is paroxysmal nocturnal hemoglobinuria (PNH) which arises from the somatic mutation of the gene encoding the phosphatidylinositol N-acetylglucosaminyltransferase subunit (*PIGA*) and results in complement-mediated hemolysis.

IBMFSs comprise a group of rare monogenic disorders characterized by blood cytopenia(s) and non-hematological effects (Table 1, Figure 1). These include Fanconi anemia (FA), dyskeratosis congenita (DC), Diamond–Blackfan anemia (DBA), Shwachman–Diamond syndrome (SDS), severe congenital neutropenia (SCN), congenital dyserythropoietic anemia (CDA), congenital amegakaryocytic thrombocytopenia (CAMT), thrombocytopenia–absent radii (TAR), and other rare entities. Two conditions have recently been described, GATA2 deficiency and *SAMD9/9L* mutations, which may be more common. Some may manifest in the neonatal period (e.g., DBA and SCN), some may develop in childhood (e.g., FA or DC), and others may present at any time in life (e.g., DC or SDS). Patients with IBMFSs do not respond to immunosuppressive therapy. The IBMFSs also behave as cancer and leukemia predisposition syndromes with a high risk of developing MDS or acute myeloid leukemia (AML) and, in some types, solid tumors. Recent advances in the genetic diagnoses of IBMFSs have identified germline mutations affecting DNA repair, telomere maintenance, and ribosome biogenesis. All these functions are necessary for the self-renewal of HSCs/HSPCs and the generation of mature blood cells. Although aplastic anemia and IBMFSs may share similar biological features of decreased hematopoietic stem, progenitor, or precursor cells, IBMFS is caused by an intrinsic defect in HSCs, whereas aplastic anemia is caused by an exogenous attack against HSCs/HSPCs. Allogeneic hematopoietic stem cell transplantation remains the only curative treatment for bone marrow failure in patients with IBMFSs or severe aplastic anemia.

There is substantial heterogeneity in the development of IBMFSs and their phenotypes even if patients share the same gene mutation. This led us to hypothesize that other factors may contribute to both hematologic and non-hematologic manifestations of IBMFSs via the production of inflammatory cytokines (Figure 2). A complex cytokine network produced by and acting on hematopoietic and stromal cells controls hematopoiesis. Dysregulation between lymphocyte and cytokine activities has been reported in aplastic anemia and hypoplastic MDS, but they have been understudied in IBMFSs. Furthermore, almost all the different forms of IBMFS are associated with an increased risk of myeloid and/or solid malignancies in which aberrant cytokine profiles may have a role. Here, we review IBMFSs, their pathogenesis, and associated cytokine abnormalities.

## 2. Common Signaling Themes

IBMFSs arise from specific alterations or abnormalities in genes associated with DNA damage repair, ribosomal structure/function, and telomere maintenance. Nevertheless, studies in patients and disease models have revealed common cytokine profiles and biological pathways underlying IBMFSs. Pro-inflammatory cytokines, IL-6 and IL-8, and an anti-inflammatory cytokine, TGF-β, were found to be commonly elevated in FA and SDS. IP-10 and IFN-γ were elevated in FA and DC (Table 2). Notably, IL-8, IP-10, and IFN-γ are elevated in patients with acquired aplastic anemia and IFN-γ has been historically implicated in its pathogenesis [2].

The shortening of telomere length, which is the hallmark of DC, is a common feature of IBMFs. Oxidative stress and reactive oxygen species (ROS) are also commonly observed in the disease models of IBMFS. Mitochondrial dysfunction commonly found in IBMFSs may further exacerbate ROS. These stress responses collectively activate the TP53/p21 axis, p16, and p38 MAPK/NF-κB axis. Exogenous stimuli such as infection, and UV and X-ray radiation add to these proinflammatory signals, resulting in the secretion of inflammatory cytokines (Figure 2). Altogether, these events lead to cell cycle arrest and apoptosis, which may explain the pathogenesis of bone marrow failure, systemic anomaly, and cancer predisposition.

Chronic expression of cytokines, e.g., IFN-α, IFN-γ, TNF-α, TGF-β, contributes to acquired bone marrow failure in inflammation-mediated mouse models [3]. The interferons signal through their cognate receptors via the JAK-STAT pathway [4], while TGF-β mediates their effects via the SMAD group of transcription factors [5]. TNF-α functions through TNFR55 and TNFR75, which activate stress-activated protein kinases [6,7]. The diversity of effects on hematopoietic stem, progenitor, precursor, and mature blood cells depends on acute versus chronic expression of these cytokines, singly or in combination, which stimulates a variety of intracellular signaling pathways.

To what extent these processes are involved in bone marrow failure development and leukemogenesis remains unclear. Levels of pro-inflammatory cytokines, such as IL-1β, IL-6, and IL-8, are elevated in serum or plasma samples obtained from patients with AML compared to age-matched healthy controls [8,9]. Leukemic stem cells and blasts isolated from AML patients release elevated levels of IL-1β and IL-6 in vitro, generating a supportive feedback loop able to maintain the proliferative capacity in vitro in the absence of the common exogenous cytokines and growth factor cocktails normally employed to stimulate cell expansion. IL-1β acts as an autocrine growth factor for AML blasts in vitro since it induces the excessive release of IL-6 and GM-CSF [10,11]. Several studies have reported elevated levels of both IL-1β and IL-1 receptors in patients with AML, whereas levels of IL-1 receptor antagonist (IL-1RA) were decreased both in peripheral blood and in bone marrow [12]. Chronic exposure to IL-1β led to alterations in the stromal niche, resulting in impaired hematopoiesis and accelerated leukemic progression in murine models [13]. Lack of IL-1RA expression inhibited AML progression in bone marrow HSPCs, partially restoring normal hematopoiesis. IL-1β regulates hematopoiesis and stem cell progenitor proliferation by p38 MAPK [8]. Since IL-1β signaling plays a key role in AML progression, multiple IL-1 blockers, including the FDA-approved canakinumab, anakinra, and rilonacept [14,15], might be beneficial in treating patients displaying preleukemic conditions, such as IBMFSs.

## 3. Common IBMFSs

### 3.1. Fanconi Anemia

#### 3.1.1. Overview

FA is perhaps the most frequent form of IBMFS and may be characterized by pancytopenia or myeloid neoplasia (MDS/AML) that often arises between 5 and 15 years of age. Systemic traits include “Fanconi” facies with microphthalmia, radial deformities, and genitourinary and other malformations. During their adolescence and young adulthood, patients are at a high risk of developing MDS/AML. Later, they are predisposed to a wide range of solid tumors, particularly esophageal/pharyngeal carcinomas, and genitourinary malignancies [16,17].

#### 3.1.2. Genetics

To date, biallelic mutations in 22 causative genes have been reported. They are involved in DNA repair and genome stability, namely the FA pathways. These are autosomal recessive (*FANCA*, *FANCC*, *FANCD1/BRCA2*, *FANCD2*, *FANCE*, *FANCF*, *FANCG/XRCC9*, *FANCI*, *FANCJ/BRIP1*, *FANCL*, *FANCM*, *FANCN/PALB2*, *FANCO/RAD51C*, *FANCP/SLX4*, *FANCQ/ERCC4*, *FANCS/BRCA1*, *FANCT/UBE2T*, *FANCU/XRCC2*, *FANCV/REV7*, and *FANCW/RFWD3*), except for *FANCB*, which exhibits X-linked recessive inheritance, and *FANCR/RAD51*, which presents a de novo autosomal dominant inheritance pattern [18]. *FANCA*, *FANCG*, and *FANCC* are the most mutated genes. A genotype–phenotype correlation may be observed in the time to develop leukemia and solid tumors in association with milder or more severe forms of mutations. *FANCD2* patients often manifest a more severe phenotype and *FANCD1/BRCA2* patients develop an early and rapidly lethal cancer-prone syndrome [19].

#### 3.1.3. Signaling Pathway

FA-associated gene products are involved in detecting interstrand crosslinking of the DNA double strands and coordinating their repair through homologous recombination. Interstrand crosslinking is caused by exogenous chemical agents such as mitomycin c and endogenous/exogenous metabolites such as aldehydes and ROS. Interstrand crosslinking prevents the separation of the DNA double helix, and unrepaired interstrand crosslinking finally causes double-strand DNA breakage, leading to nonhomologous end-joining of double-strand DNA breakage. Interstrand crosslinking can be solely properly repaired via homologous recombination. The FA core complex composed of three assemblies (FANCA-FANCG-FAAP20, FANCB-FANCL-FAAP100, and FANCC-FANCE-FANCF) is recruited at interstrand crosslinking and double-strand DNA breakage by recognition of the anchoring complex (FANCM, FAAP24, and MHF1/2) [20]. The core complex is an E3 ligase that monoubiquitinates and activates the heterodimer, FANCD2 and FANCI, which further recruits the downstream proteins. FANCQ/XPF makes incisions around the lesion (called “unhooking”) [21], which is dependent on FANCP/SLX4. Translesion polymerase, Polζ, which includes FANCV/REV7 [22], creates a substrate for homologous recombination. FANCD1/BRCA2, FANCJ/BRIP1, FANCN/PALB2, FANCO/RAD51C, FANCR/RAD51, FANCS/BRCA1, FANCU/XRCC2, and FANCW/RFWD3 are proteins that regulate homologous recombination. Once the interstrand crosslinking repair is completed, the pathway is turned off by a deubiquitinating enzyme, USP1. These comprise the canonical FA pathways.

The characterization of signaling pathways in FA cells identified a role for the hyperactive TGF-β pathway [23]. This non-canonical pathway suppresses the survival of FA cells upon DNA damage, ultimately leading to bone marrow failure. Thereafter, senescence pathways have been identified to expand this non-canonical pathway. FA cells are hypersensitive to endogenous and exogenous stresses, leading to the unrestricted activation of DNA damage response (hyperactive ATM/TP53/p21 pathway, ATR/CHK1 pathway, p16/RB, NF-κB, and p38) and cell cycle arrest. The impairment of DNA damage repair, mitochondrial dysfunction, elevated cellular ROS, and senescence-associated secretory phenotype factors (e.g., TNF-α and TGF-β) further exacerbate the stress pathway. Oxidative stress leads to progressive TP53-dependent depletion of the HSC/HSPC pool. The hallmarks of senescence have been confirmed, including the expression of senescence-associated heterochromatin foci and SA-β-gal in FA cells [24]. Recently, FANC proteins have been shown to affect nucleolar homeostasis and ribosome biogenesis, further expanding “non-canonical” pathways in FA. FA proteins protect the nascent DNA strands when replication is stalled [25]. They function at sites of under-replicated DNA, known as common fragile sites, and have been shown to play a role in the clearance of DNA:RNA hybrids, which form during transcription and are enhanced by replication and transcription machinery collisions [26].

#### 3.1.4. Inflammatory Profile

Patients with FA have been tested for immunological and inflammatory profiles. Patients with FA show decreased number of B cell lymphocytes and NK cells compared to normal controls, and impaired function in cytotoxic T lymphocytes. Immunoglobulin levels are variable among FA patients; however, patients with FA who developed severe bone marrow failure showed decreased levels of IgG and IgM [27]. Although there is no consensus on cytokine profiles in patients with FA, increased levels of serum TGF-β, IL-6, and low soluble CD40L compared to healthy controls were reported. The levels of IL-1β, IL-2, IL-4, IL-10, IL-13, IL-17, and IL-23 were not different in this study [28]. In another report, higher plasma levels of IL-10 in FA patients but no difference in TGF-β were noted [29]. TNF-α and IFN-γ have been proposed as causative stress in bone marrow failure in aplastic anemia [30]. These inflammatory cytokines may play a role in enhancing oxidative stress and DNA damage in FA pathogenesis. TNF-α and IFN-γ were significantly overexpressed in stimulated mononucleated cells from the bone marrow of FA patients as compared to healthy controls [31]; however, this was not observed in another cohort [32]. T cell lymphocytes from FA patients showed increased expression of TNF-α and IFN-γ in one report; however, this was not observed in the other, which showed an increased tendency of peripheral monocytes to produce TNF-α, IL-6, and IL-1β in response to low dose lipopolysaccharide [33]. *FANCA* patients showed elevated levels of IL-1β due to the constitutive activation of the PI3K-AKT pathway [34]. Lymphoblastoid cell lines established from *FANCA* and *FANCC* patients exhibited the overexpression of secretory factors including IL-6, IL-8, MMP-2, and MMP-9 compared to control cells. Knockdown of *FANCA* or *FANCC* in MDA-MB-231 breast cancer cell lines using siRNA induced NF-κB-dependent expression of IL-6, IL-8, MMP-2, and MMP-9 [35]. Still, the contributions of inflammatory cytokines to FA pathogenesis remain to be fully determined [36,37]. Inhibition of TGF-β by luspatercept, a trap for the TGF family of ligands [38], may be of clinical value for the anemia of FA.

### 3.2. Dyskeratosis Congenita

#### 3.2.1. Overview

Increasingly referred to as short telomere syndrome, DC is characterized by the triad of abnormal skin pigmentation (reticular hyperpigmentation), oral leukoplakia, and nail dystrophy (punctate leukonychia pitting, leading to shedding of nails). Other common and diagnostic symptoms include learning difficulties/developmental delay, short stature/intrauterine growth restriction, pulmonary disease, dental caries/loss, esophageal stricture, premature hair loss/greying, cancer, liver disease, ataxia/cerebellar hypoplasia, microcephaly, and osteoporosis [39]. Nail changes may be the first presentation of the disease [40]. Later in the course, patients develop bone marrow failure, pulmonary fibrosis, and/or cancer [41]. Bone marrow failure develops in up to 80% of patients by the age of 30 [39]. Patients with DC are predisposed to develop MDS/AML and squamous cell carcinomas of the head and neck. The prevalence of DC in the general population is estimated at nearly one in a million [42].

There is considerable heterogeneity among patients with DC regarding the onset and severity of the symptoms, even among related individuals. Overlapping disorders exist. Hoyeraal–Hreidarsson syndrome is characterized by intrauterine growth restriction, microcephaly, cerebellar hypoplasia, and variable immune deficiency, which typically presents in infancy, as the patients progressively develop bone marrow failure. Revesz syndrome is a very rare syndrome characterized by bilateral exudative retinopathy in infancy and early-onset bone marrow failure. Common characteristics are cerebellar hypoplasia, cerebral calcifications, neurodevelopmental delay, and other nonhematological symptoms observed in DC patients [43]. Coats plus syndrome is another very rare syndrome characterized by retinal telangiectasia and exudates, intracranial calcification with leukoencephalopathy and brain cysts in early childhood, gastrointestinal bleeding (due to intestinal vascular ectasia), osteopenia, bone marrow failure, and other DC-related symptoms, which overlap with Revesz syndrome [44]. These syndromes constitute a severe form of DC.

#### 3.2.2. Genetics

Patients with DC and DC-like diseases have short telomeres compared to age-matched controls, and DC is recognized as a telomeropathy, whose telomere maintenance is defective, resulting in significantly short telomeres, affecting genome stability and limiting proliferative lifespan. DC-causative genes have been identified (18 genes, to date) and these account for three-quarters of cases [39]. To mitigate the shortening of telomeres in dividing cells, telomerase is recruited. Telomerase is a ribonucleoprotein complex composed of telomerase reverse transcriptase (encoded by *TERT*) and the non-coding RNA, hTR, serving as a template for elongating telomeres (encoded by *TERC*) [45]. The dyskerin complex consisting of dyskerin (encoded by *DKC1* on the X chromosome, the most frequently mutated gene in DC), NOP10 (*NOP10*), and NHP2 (*NHP2*) binds to the H/ACA domain of hTR. This domain also contains the CAB box, which binds the telomerase Cajal body protein 1, TCAB1 (*WRAP53*), for trafficking telomerase to Cajal bodies. NAF1 (*NAF1*) binds to dyskerin and is required for the stable association of dyskerin with telomerase. Stably associated holoenzyme components are required for functioning telomerase. Shelterin is comprised of up to six different proteins including TIN2 (encoded by *TINF2*, the second most mutated gene in DC), ACD (or TPP1 encoded by *ACD*), and POT1 (*POT1*). This complex binds both the single-stranded and double-stranded DNA regions of telomeres which have roles in both telomere protection and telomerase regulation. The single-stranded 3′ tail invades the double-stranded DNA region to form a displacement loop (D-loop) and a telomere loop (T-loop). T-loop formation is regulated by shelterin and can restrict telomerase access to the 3′ tail. Once the telomere has opened up, telomerase can bind to the 3′ tail through its RNA template and add telomeric repeats. The CST complex, comprising CTC1 (*CTC1*), STN1 (*STN1*), and TEN1, assists DNA replication at telomeres and then inhibits telomerase activity [46].

Other protein-coding genes for the telomere maintenance machinery have been identified in patients with DC and DC-like diseases. These include *ZCCHC8*, encoding a scaffold subunit of a nuclear exosome targeting component required for telomerase RNA maturation/degradation; *RPA1*, encoding a DNA replication protein complex that binds to single-stranded DNA, which is required for telomere maintenance; *PARN*, encoding a poly(A)-specific ribonuclease which regulates telomerase and shelterin-composing transcripts via regulating *TP53* expression [47]; and *DCLRE1B*, encoding a repair exonuclease interacting with TRF2 in shelterin to protect telomeres. Biallelic mutations of *RTEL1* cause very short telomeres. *RTEL1* encodes a DNA helicase that has roles in T-loop unwinding, regulating DNA replication and DNA recombination, promoting the mitotic DNA synthesis pathway, influencing RNA trafficking, and regulating telomere-repeat-containing RNAs [48].

#### 3.2.3. Signaling Pathway

Very short telomeres induce replicative senescence and the ability to protect chromosomal ends. These will be recognized as damaged DNA which recruits cellular senescence or apoptosis pathways via the activation of TP53 [49]. Fibroblasts from DC patients showed phosphorylated TP53 and upregulated expression of *CDKN1A*, which was accompanied by elevated oxidative stress markers. The poor proliferation of DC cells was partially overcome by reducing oxygen tension [50]. Interestingly, a patient with a DC-like disorder harbored a germline missense mutation in *MDM4* without known DC-causing gene mutations. The patient had a history of neutropenia, hypocellular bone marrow, and vague gastrointestinal symptoms. The proband’s mother and cousin harboring the same mutation had intermittent neutropenia and hypocellular bone marrow. The lymphocyte telomeres were between the 1st and 10th percentiles in the proband and the cousin. The mutation caused lower expression of MDM4, a negative regulator of TP53, leading to lower TP53 proteins and short telomeres [51]. Together with PARN and WRAP53 (WD repeat containing antisense to TP53), activation of TP53 is established in the pathogenesis of telomeropathies.

#### 3.2.4. Inflammatory Profile

Immune defects can occur in patients with DC, especially in severe forms [39]. There is a reduction in the B cell lymphocytes and NK cells, whereas T cells are relatively spared and maintain normal function [52]. Immunodeficiency is seen in virtually all patients with biallelic variants in *RTEL1*, while immunological phenotypes are more variable in patients with other gene mutations [53,54]. Loss of the HSC pool can also result in decreased circulating levels of B and T cells and monocytes [33]. However, immunological abnormalities may occur in the absence of profound bone marrow failure.

Several reports document immunological and inflammatory changes in patients with DC. G-CSF, Flt3L, and CXCL10 (IP-10) were increased in the sera from DC patients who developed severe bone marrow failure, whereas RANTES was lower than in DC patients with mild to moderate bone marrow failure or healthy subjects [32,37]. In a male with a severe form of DC harboring a hemizygous mutation in *DKC1* (p.R449G), single-cell RNA-seq revealed that IFN-related genes (*IFNAR1*, *JAK1/2*, *TYK2*, and *STAT1/2*), interferon-stimulated genes, and interferon-inducible genes were upregulated, while *IFNG*, genes involved in IFN-γ expression, NF-κB genes, and genes involved in NLRP3 inflammasome formation were all reduced in peripheral blood mononuclear cells, suggesting an impairment of pro-inflammatory cytokine production and secretion in the patient [55].

### 3.3. Diamond–Blackfan Anemia

#### 3.3.1. Overview

DBA presents as severe macrocytic anemia in neonates and infants. In 90% of patients, anemia starts before 12 months of age [56]. Patients with DBA also manifest craniofacial anomalies in 50% of cases, growth delay in 30%, and various abnormalities of the limbs (especially the thumbs, in approximately 40%) and the viscera (such as the genitourinary and cardiac systems). Bone marrow shows pure red cell aplasia at the level of erythroid precursors but the other lineages are spared, although a study suggested that myeloid and lymphoid precursors are also affected, leading to neutropenia and lymphopenia [57]. DBA is associated with an increased risk of MDS, AML, and solid tumors, including osteosarcoma and colon carcinoma [58].

#### 3.3.2. Genetics

DBA is the first disease described as a ribosomopathy. Mutations, deletions, and copy number changes have been identified in 20 of the 80 ribosomal protein genes [59]. These include genes encoding the small (*RPS7*, *RPS10*, *RPS15A*, *RPS17*, *RPS19* (the most frequent, 25%), *RPS24* (2–3%), *RPS26* (7–9%), *RPS27*, *RPS28*, and *RPS29*) and the large (*RPL5* (the second most frequent, 7–12%), *RPL9*, *RPL11* (5–7%), *RPL15*, *RPL18*, *RPL26*, *RPL27*, *RPL31*, *RPL35*, and *RPL35A* (2–3%)) ribosomal subunits. Non-ribosomal protein genes, *GATA1* and *TSR2,* were reported to be causative genes of DBA, although there is a discussion on whether these are classical DBA or DBA-like diseases [56]. TSR2 is involved in the pre-rRNA processing and binds to RPS26 [60]. GATA1 is the major erythroid transcription factor and plays a critical role in regulating normal erythroid differentiation. *GATA1* germline mutations may lead to GATA1-related cytopenia in males, which is characterized by thrombocytopenia and/or anemia ranging from mild to severe, and one or more of the following: platelet dysfunction, mild β-thalassemia, neutropenia, and congenital erythropoietic porphyria [59]. However, the mechanism whereby a defect in ribosomal proteins leads to a specific defect in erythropoiesis has not been fully understood. HSP70 and ribonuclease inhibitor 1 that binds to the 40S ribosome small subunit may be involved in the translational control of GATA1, which affects erythropoiesis. A global reduction in ribosome levels in DBA with a normal ribosome composition altered the translation of specific RNA transcripts, affecting lineage commitment in hematopoiesis [61].

#### 3.3.3. Signaling Pathway

Association of DBA with TP53 has been reported. Depleting RPS19 or RPL11 in CD34^+^ cord-blood-derived erythroid progenitors induced TP53 activation and its target genes (*CDKN1A*, *BAX*, and *PMAIP1*) [62]. Ribosomal proteins (RPS3, RPS7, RPS27, RPS27a, RPL5, RPL11, and RPL23) directly bind MDM2, a negative regulator of TP53. Among them, excess RPL5 and RPL11 activate TP53 as both a sensor and effector of ribosomal stress [63]. GATA1 has been reported to interact directly with and inhibit TP53 [64]. ROS that result from an imbalance between decreased globin synthesis and excess of free heme could induce apoptosis in erythroid progenitors and precursors. Autophagy and cell metabolism may also have a role in DBA pathophysiology [65].

#### 3.3.4. Inflammatory Profile

In contrast to the other IBMFSs, no significant changes in pro-inflammatory cytokines (e.g., TNF-α and IFN-γ) have been noted in DBA patients [32,33]. In a study comparing serum cytokines in patients with IBMFSs, only patients developing severe bone marrow failure in FA and DC showed high serum levels of G-CSF and Flt3L and low levels of RANTES. Patients with DBA had no elevation of cytokines [32]. Peripheral lymphocytes and monocytes are lower in DBA patients compared with controls. After stimulation with phorbol 12-myristate 13-acetate and ionomycin, TNF-α and IFN-γ production by CD3^+^ T cells is decreased in DBA compared with healthy subjects and other IBMFS, as well as that of TNF-α-producing CD14^+^ monocytes [33]. The serum soluble form of FasL (sFasL) was significantly elevated in patients with DBA compared to age-matched healthy controls, but serum IFN-γ was not [66]. Nevertheless, a certain number of patients respond to glucocorticoid treatment in DBA [67]. While the mechanism for steroid responsiveness is not known, steroids are potent inhibitors of cytokine production. A recent single-cell RNA-seq analysis of erythroid progenitors isolated from the bone marrow of DBA patients without glucocorticoid treatment showed high expression of the G1/S transition gene set of the cell cycle, suggesting that erythroid progenitors are forced to progress into the cell cycle rather than cell cycle arrest. Glucocorticoids reduce free ribosomal proteins that elicit nucleolar stress and attenuated cell cycle progression via elevating IFN signaling. Notably, IFN-α treatment in vitro could be an alternative to glucocorticoid treatment or even an add-on effect to steroids in erythroid differentiation [68]. The role of IFN-α in IBMFSs will be discussed in the section on CDA.

### 3.4. Shwachman–Diamond Syndrome

#### 3.4.1. Overview and Genetics

SDS is one of the most common IBMFSs, with an incidence of 1:75,000–1:168,000 [69,70]. More than 90% of patients diagnosed with SDS carry biallelic mutations in the Shwachman–Bodian–Diamond Syndrome (*SBDS*) gene. However, other genes have been recently associated with SDS or SDS-like conditions, including DnaJ heat shock protein family (Hsp40) member C21 (*DNAJC21*), signal recognition particle 54 (*SRP54*), and elongation factor-like GTPase 1 (*EFL1*), as we recently reviewed [71]. Since SBDS, EFL1, DNAJC21, and SRP54 are all involved in ribosome biogenesis or affect the total protein synthesis, SDS has been classified as a ribosomopathy. SBDS physically interacts with the GTPase EFL1 to promote the release of eukaryotic translation initiation factor 6 (EIF6) from the pre-60S subunit, allowing the proper temporal and spatial assembly of the eukaryotic 80S ribosome from the 40S SSU (small subunit) and the 60S LSU (large subunit) [72,73]. SRP54 is a component of the signal recognition particle (SRP) ribonucleoprotein complex, involved in the co-translational targeting of proteins to the endoplasmic reticulum [74], whereas DNAJC21 is involved in 60S ribosomal subunit maturation.

SDS is clinically characterized by exocrine pancreas insufficiency, skeletal abnormalities, short stature, and bone marrow failure. In SDS bone marrow, myeloid progenitor differentiation is arrested at the myelocyte–metamyelocyte stage [75]. Patient-derived CD34^+^ HSCs showed increased apoptosis, due to dysregulated Fas–Fas ligand signaling [76]. This results in a hypocellular bone marrow, particularly in the myeloid lineage. Most patients show moderate to severe neutropenia early in life. Anemia and thrombocytopenia are less frequent [77]. However, data from the Italian cohort of patients indicated that the number of B lymphocyte cells and some subsets of T lymphocyte cells, including double-negative T lymphocyte cells, is decreased in SDS [77]. Similar to other IBMFSs, SDS is associated with a high-risk (~500-fold compared to age-matched controls) of transforming into MDS/AML [78].

Almost 50% of patients from an English cohort of SDS patients reported duodenal inflammatory features. Immunohistochemical analysis revealed an increased number of inflammatory cells, including lymphocytes, macrophages, eosinophils, and CD20^+^ B cells, without an active neutrophilic component, mainly localized in the deep lamina propria around crypt bases [79].

#### 3.4.2. Signaling Pathway

Knockdown of *SBDS* by short hairpin RNA in cervical cancer HeLa cells and TF-1 myeloid cells showed that decreased *SBDS* expression is associated with increased release of ROS. The increased oxidative stress led to accelerated Fas-mediated apoptosis and reduced cell growth, which were partially rescued by anti-oxidants such as N-acetylcysteine [80]. Cre-mediated deletion of *Sbds* from osterix^+^ mesenchymal progenitor cells led to disruption of the cortical bone and marrow architecture in a mouse model recapitulating SDS skeletal abnormalities [81]. Using the same murine model, the investigators found that HSCs/HSPCs display mitochondrial dysfunction including mitochondrial hyperpolarization associated with a marked increase in intracellular ROS levels and DNA damage, with nuclear foci enriched in Ser139-phosphorylated H2AX histone (γH2AX). In addition, HSPCs from the mutant mice showed increased *Trp53* and *Cdkn1a*, suggesting cellular senescence [82]. These data were consistent with a previous study conducted on *SBDS*-depleted HEK293 cell lines showing hypersensitivity to DNA damage and UV irradiation [83]. Moreover, targeted depletion of *Sbds* in the murine pancreas resulted in an early p53 stabilization in acinar cells, already during the postnatal period. Senescent acinar cells showed upregulation of NF-kB transcription factor and increased release of TGF-β [84].

#### 3.4.3. Inflammatory Profile

Elevated plasma levels of TGF-β were found in SDS patients and upregulation of the TGF-β pathway was observed in SDS HSPCs. TGF-β pathway activation through TGFβR1 suppressed hematopoiesis in normal HSPCs in vitro. On the contrary, in vitro treatment with TGF inhibitors, including AVID200 and SD208, improved hematopoiesis in SDS HSPCs [85]. Several inflammatory conditions have been reported in patients with SDS, including juvenile idiopathic arthritis, chronic recurrent multifocal osteomyelitis, and scleroderma. These patients reported elevated levels of pro-inflammatory chemokines belonging to the NF-κB pathway, such as IL-8, and the chemokine family including CCL16 and CCL21 [86].

We reported that the signal transducer and activator of transcription 3 (STAT3) transcription factor is hyperphosphorylated in SDS leukocytes [87]. The mammalian target of rapamycin (mTOR) pathway seems to play a key role in STAT3 hyperactivation in SDS [87,88]. Consistent with STAT3 hyperactivation, we found that lymphocytes and bone marrow nuclear cells from SDS patients released elevated levels of the pro-inflammatory cytokine IL-6 in vitro. Patients with SDS showed in vivo elevated plasma levels of IL-6 both in peripheral blood and bone marrow [89].

## 4. Other Rare IBMFSs

SCN is characterized by impaired production of neutrophil granulocytes and patients often develop life-threatening infections in their neonatal period. Gingivitis is common with inflammatory cytokines found in the inflammatory exudate derived from the periodontal tissues. The most common cause is mutations in *ELANE* (~50%), followed by *HAX1* and *G6PC3* mutations (10–20%) [90]. Patients with SCN are predisposed to MDS/AML, which are associated with somatic mutations in *CSF3R* and *RUNX1*. G-CSF therapy has been linked to an elevated risk of myeloid malignancies. *ELANE* mutations correlated with more severe periodontal status than other genotypes, having higher levels of pro-inflammatory IL-1β in gingival crevicular fluid [91]. Another report demonstrated that chemokines (IP-10, MIG, and MIP-1β), and pro-inflammatory (TNF-α, IL-2, IL-7, IL-15, IL-17, and IL-33) and anti-inflammatory cytokines (IFN-α, IL-10, and IL-13) were significantly lower in the patients with SCN, compared to the healthy controls, which likely correlated with neutrophil deficiency [92].

CDA is characterized by inefficient erythropoiesis resulting in anemia [93]. CDA was historically classified into three major types (I, II, III) based on the morphological features of erythroblasts in the bone marrow. Clinical and genetic studies added CDA IV and CDA-like syndromes. CDA I is caused by biallelic mutations of *CDAN1* or *CDIN1*, CDA II (the most common CDA) by *SEC23B*, CDA III by a heterozygous mutation of *KIF23*, and CDA IV by a heterozygous mutation of *KLF1* [94]. Although the precise mechanism leading to erythroid cytopenia has not been uncovered, CDIN1, KIF23, and SEC23B are involved in cytokinesis. Causative genes in CDA-like syndromes include *GATA1*, *ALAS2*, *LPIN2*, *CAD*, *COX4I2*, *MVK*, *PARP4*, *VPS4A*, and *PRDX2* in a single case report or few case series [93]. These need to be validated. While inflammatory cytokines have not yet been reported in patients with CDA, a few reports have demonstrated the success of IFN-α in rescuing anemia. Based on a case report where an adult patient with CDA I treated with IFN-α2a for chronic hepatitis C due to repeated transfusion showed an increased hemoglobin level to the normal range, and that discontinuation resulted in returning to previous values [95], IFN-α treatment was tested in a few patients with CDA with controversial results. All six children with CDA administered varying dosages and frequencies of IFN-α2b showed no favorable effect on hemoglobin, reticulocyte count, or transfusion frequency [96]; however, an adult patient with CDA receiving IFN-α2a achieved a normal range of hemoglobin with markedly reduced CDA-specific dysplasia [97], and an adolescent and an adult patient with CDA I and β thalassemia carriers received IFN-α2b, reducing their transfusion requirements [98].

Germline *SAMD9* and *SAMD9L* mutations are associated with SAMD9/SAMD9L syndrome with a clinical spectrum of disorders including MIRAGE (myelodysplasia, infection, restriction of growth, adrenal hypoplasia, genital problems, and enteropathy) syndrome [99], ataxia–pancytopenia syndrome [100], and myelodysplasia and leukemia syndrome with monosomy 7 [101]. Patients with SAMD9/SAMD9L syndrome present with heterogeneous clinical manifestations: *SAMD9* mutations appear to be associated with a more severe disease phenotype, including intrauterine growth restriction, developmental delay, and organ hypoplasia, whereas *SAMD9L* mutations have been more often linked to ataxia due to cerebellar atrophy. Blood disorders vary from mild and transient cytopenia with dysmorphic changes to rapid progression to MDS/AML with monosomy 7. Insights into SAMD9/SAMD9L functions have been obtained from studies on genetic reversion in human patients. Germline *SAMD9*/*SAMD9L* mutations are gain-of-function mutations that cause pancytopenia and generally restricted growth and/or specific organ hypoplasia in non-hematopoietic tissues [101]. SAMD9 and SAMD9L colocalize with EEA1, which promotes the homotypic fusion of endosomes and degradation of receptor proteins, and they interfere with endosomal PDGFRβ, thereby downregulating its downstream signaling in human cells [102]. SAMD9L is highly expressed in NK cells and monocytes, presenting with low numbers of NK cells and monocytes in patients with SAMD9/SAMD9L syndrome. IFN-α or IFN-γ induced SAMD9L expression in peripheral-blood-derived NK cells, bone-marrow-derived CD34^+^ HSCs, and fibroblasts from healthy humans [103]. In *Samd9l* mouse models, *Samd9l* heterozygous knockout predisposes mice to myeloid malignancies. *Samd9l* plays an important role in the degradation of cytokine receptors by endocytosis and endosome fusion with lysosomes in this model [102]. *Samd9l* biallelic gain-of-function mutants developed bone marrow failure, growth retardation, and both homozygous and heterozygous mutant mice show reduced repopulating capacity [104]. Inflammation induced by pI:pC reduced the engraftment potential of both *Samd9l*-WT and conditional knockin *Samd9l*-W1171R mutant (*Samd9l*-Mut), but it increased the apoptosis of bone marrow cells, leading to hypocellularity in *Samd9l*-Mut, as opposed to WT. Upregulation of TGF-β pathways with increased p-SMAD2/3 has been shown to lead to HSPC exhaustion [105,106].

GATA2 deficiency is caused by germline mutations in the *GATA2* gene. Clinical features include nontuberculous mycobacterial, bacterial, fungal, and human papillomavirus infections, lymphedema, pulmonary alveolar proteinosis, and bone marrow failure, typically presenting in adolescents. Patients with GATA2 deficiency are predisposed to develop MDS/AML [107]. GATA2 is a transcription factor that is critical for embryonic development, maintenance, and proliferation/maintenance of HSCs/HSPCs. Single-cell RNA-seq of HSPCs from patients with GATA2 deficiency revealed downregulated genes highly enriched in immune responses (immune system, infectious disease, and cytokine signaling) and cell cycle and proliferation [108]. Its deficiency leads to hypocellular bone marrow and a decrease in or absence of monocytes, B cell precursors, B cell NK cells, or plasmacytoid dendritic cells [109]. GATA2 deficiency has been associated with elevated serum Flt3L. A proteomic screen revealed trends for increased FGF-2, EGF, GM-CSF, and CD40L in patients with GATA2 deficiency compared with healthy controls [110].

MECOM (*MDS1* and *EVI1* complex locus on 3q26.2)-associated syndromes have been associated with CAMT and radioulnar synostosis. *MECOM* haploinsufficiency is recognized to be a cause of severe neonatal bone marrow failure with near-complete loss of HSCs. The hematological defects range from B cell deficiency to pancytopenia, and various systemic manifestations are observed (clinodactyly, presenile hearing loss, and cardiac/renal malformations) [111]. MECOM overexpression has been found in adult and childhood AML with a poor prognosis. *MECOM* editing in human CD34^+^ HSPCs led to a reduction in LT-HSCs, and CFU-GEMM and CFU-GM were decreased while more differentiated CFU-G and CFU-M were increased in MethoCult H4034. Single-cell genomic analyses revealed that CTCF, a regulator of genome organization anchoring cohesin-based chromatin loops, mediates dysregulation of HSC quiescence by MECOM. CTCF occupancy was highly conserved across erythroid cells, T cells, B cells, and monocytes [112]. As a critical regulator of hematopoiesis, MECOM inhibits TGF-β by interacting with SMAD3, and stress-induced cell death by inhibiting JNK [113,114].

## 5. The Role of Bone Vasculature in Bone Marrow Inflammation

The vasculature is involved in the maintenance and proliferation of HSCs/HSPCs in bones and the release of mature cells and platelets into the peripheral circulation, constituting an important component of the bone marrow microenvironment. The vasculature regulates the differentiation of perivascular mesenchymal stromal cells into bone cells. The release of stromal-cell-derived factor 1 (CXCL12) from proliferating lymphatic endothelial cells provides a driving force during HSC/HSPC proliferation and bone regeneration. Lymphangiogenesis in bones is regulated by IL-6 through VEGF-C/VEGFR-3 signaling and genotoxic stress [115]. Additionally, the vasculature plays an important role in the inflammation of the bone marrow. Bone marrow endothelial cells constitutively release cytokines and growth factors, including IL-6, Kit-ligand, GM-CSF, and G-CSF, supporting the long-term proliferation and differentiation of HSPC [116]. Elevated plasma levels of IL-1β and TNF-α induced by infections may stimulate endothelial cells to upregulate cytokine release and adhesion molecules [117,118], amplifying the innate immune response. The effect may be paradoxical, with the inflammatory process being essential to containing infections but also impairing hematopoiesis by reducing the differentiation and proliferation of progenitor cells in the bone marrow. Using transgenic mice, Fernandez and colleagues reported that bone marrow endothelial cells regulate the proliferation of HSPCs via a Notch-dependent mechanism that may be triggered upon TNF-α and lipopolysaccharide stimulation [119].

Interestingly, bone marrow mesenchymal stromal cells isolated from patients with SDS show defective angiogenesis, resulting in reduced networks, with impaired capillary tubes and vessels and significantly reduced VEGFA expression [120]. It has also been observed that angiogenesis in SDS bone marrow biopsy specimens is increased compared with normal bone marrows, resulting in improved marrow microvessel density and normal levels of serum VEGF [121]. Further studies are needed to clarify whether the vasculature is impaired in SDS. Little is known about the bone marrow vasculature in other IBMFSs. In a cohort of 18 patients with acquired aplastic anemia, VEGF expression and microvessel density in bone marrow were significantly reduced in these patients compared to healthy donors [122].

**Table 2 biomolecules-13-01249-t002:** Summary of inflammatory profiles in IBMFSs.

	FA	DC	DBA	SDS
Anti-inflammatory				
TGF-β	↑ *			↑
IL-10	↑			
sFasL			↑	
Pro-inflammatory				
IL-1β	↑ *			
IL-6	↑ *			↑
IL-8	↑			↑
TNF-α	↑ *		↓	
IFN-γ	↑ *	↑ *	↓	
sCD40L	↓			
RANTES	↓	↓		
CXCL10 (IP-10)	↑	↑		
CCL16				↑
CCL21				↑
G-CSF	↑	↑		
Flt3L	↑	↑		

This table was produced using data collected from refs [28,29,31,32,33,37,66,85,86,89]. Abbreviations: FA, Fanconi anemia; DC, dyskeratosis congenita; DBA, Diamond–Blackfan anemia; SDS, Shwachman–Diamond syndrome. ↑, increased levels; ↓, decreased levels; * Conflicting data exist.

## 6. Conclusions

A variety of genes cause IBMFSs via canonical signal pathways underlying DNA damage repair, ribosomal structure/function, and telomere maintenance. Crosstalk exists between oxidative stress, DNA damage, telomere shortening, and ribosomal dysfunction. These stresses elicit TP53 responses, but various experimental studies suggest a role for inflammation. Inflammatory cytokine profiles (e.g., IL-6, IFN-γ, and TGF-β) are commonly observed in patients with FA, DC, DBA, and SDS. Together with TP53 activation, these cytokines result in growth arrest, recruitment of p16/p21-mediated senescence, and apoptosis. Cytokine signatures provide clues for a better understanding of the pathogenesis of IBMFSs. Additional studies may identify biomarkers for specific genotypes of IBMFSs, thereby providing better surveillance of IBMFSs. These studies may lead to the use of anti-inflammatory drugs or the development of new agents to reduce the severity of bone marrow failure. These studies might establish that a chronic release of pro-inflammatory cytokines might drive not only the cytopenias but also the transformation to cancer and leukemia.

## Figures and Tables

**Figure 1 biomolecules-13-01249-f001:**
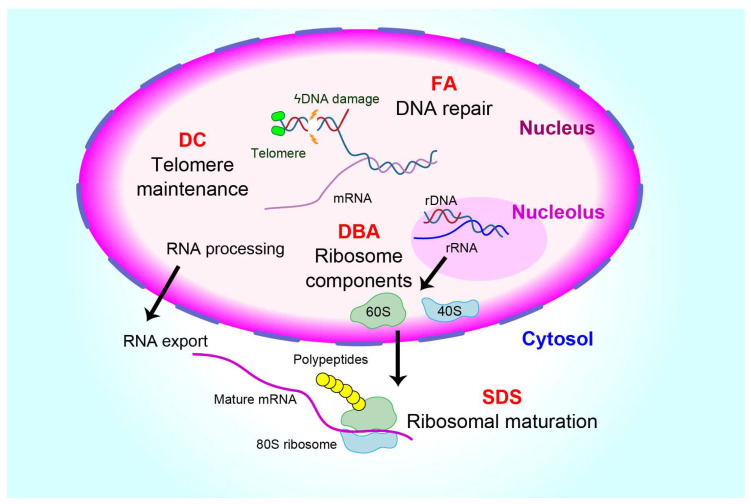
Landscape of genetic alterations in inherited bone marrow failure syndromes (IBMFSs). The most common IBMFSs are Fanconi anemia (FA), Diamond–Blackfan anemia (DBA), dyskeratosis congenita (DC), and Shwachman–Diamond syndrome (SDS). FA-causing genes encode proteins responsible for DNA repair. DC-causing genes encode proteins for maintaining telomere length. Most mutated genes in patients with DBA encode ribosomal proteins, while SDS results from genes regulating the assembly of the ribosome. Common to these diverse IBMFSs is the activation of stress pathways, which include TP53-dependent and senescence-associated cytokine responses.

**Figure 2 biomolecules-13-01249-f002:**
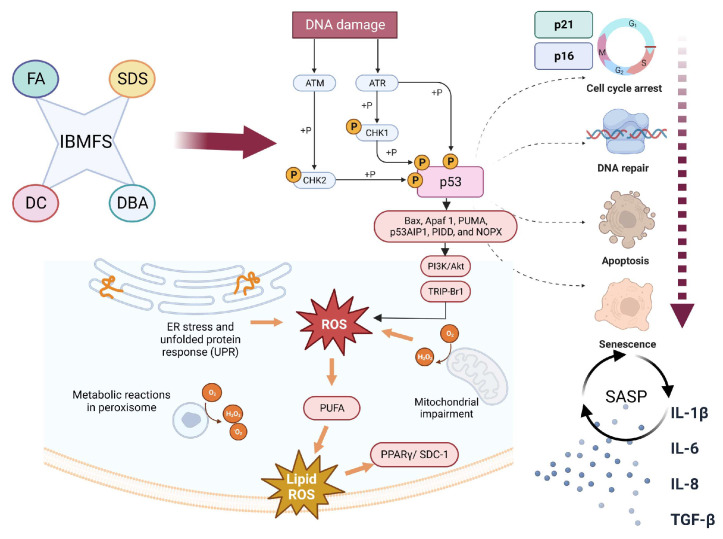
Senescence pathways in inherited bone marrow failure syndromes (IBMFSs). Hematopoietic stem, progenitor, and precursor cells undergo a variety of stresses when genes involved in DNA damage repair, telomere maintenance, and protein synthesis are mutated. These stress responses include activation of the TP53/p21 (CDKN1A) axis, p16, and p38 MAPK/NF-κB axis. In IBMFSs, these pathways may be chronically activated with occasional spikes due to additional environmental stimuli such as infections, and result in the secretion of inflammatory cytokines (e.g., IL-1β, IL-6, IL-8, and TGF-β). Mitochondrial dysfunction and unfolded protein response (UPR) lead to elevated reactive oxygen species (ROS), further exacerbating stress responses, and leading to stress pathway activation that may be collectively referred to as senescence. Altogether, these events impair cell cycle arrest and apoptosis. Abbreviations: FA, Fanconi anemia; DC, dyskeratosis congenita; DBA, Diamond–Blackfan anemia; SDS, Shwachman–Diamond syndrome; ER, endoplasmic reticulum; SASP, senescence-associated secretory phenotype; PUFA, polyunsaturated fatty acids.

**Table 1 biomolecules-13-01249-t001:** Clinical characteristics of IBMFSs.

	FA	DC	DBA	SDS	SCN
Estimated incidence (/1,000,000 births/year) *	11.4	3.8	10.4	8.5	4.7
Inheritance pattern	AR > AD, XLR	XLR > AR	AR > AD	AR	AD
Typical hematological findings	Pancytopenia	Pancytopenia	Anemia	Neutropenia	Neutropenia
Extrahematological symptoms	Growth deficiency, skeletal anomalies (radial axis), skin pigmentation, small head/eyes, genitourinary anomaly, reproductive problems	Growth deficiency, abnormal nails, leukoplakia, reticular pigmentation, pulmonary fibrosis, grey hair, cerebellar hypoplasia	Growth deficiency, head/facial anomaly, skeletal anomaly (thumb), kidney/heart anomaly	Growth deficiency, skeletal anomaly (metaphyseal dysostosis), exocrine pancreatic insufficiency	Not common
Short telomere	+	++	+	+	−
MDS/AML predisposition	+	+	+	+	+
Solid cancer predisposition	Squamous cell carcinoma (head/neck, genitourinary)	Squamous cell carcinoma (head/neck, skin)	Osteosarcoma	Not common	Not common

* Data from ref. [1]. Abbreviations: FA, Fanconi anemia; DC, dyskeratosis congenita; DBA, Diamond–Blackfan anemia; SDS, Shwachman–Diamond syndrome; SCN, severe congenital neutropenia; AR, autosomal recessive; AD, autosomal dominant; XLR, X-linked recessive; MDS, myelodysplastic syndrome; AML, acute myeloid leukemia. +, sometimes present; ++, almost always present; −, mostly absent.

## Data Availability

The data presented in this study are available in this article and the references.

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
