# Peer review of "The Molecular and Genetic Mechanisms of Inherited Bone Marrow Failure Syndromes: The Role of Inflammatory Cytokines in Their Pathogenesis"

_biomolecules, 2023, doi:10.3390/biom13081249_

Round 1

Reviewer 1 Report

 The review is summarizing the pathological mechanisms of various types of inherited bone marrow failure syndromes (IBMFS). The review is broad, includes a lot of data but some chapters are presented chaotically, which hardens to follow up this review. The manuscript needs better structuration and more clear presentation of data.

1.      The title does not match the whole review, i.e. it says that everything is about the role of inflammatory cytokines in the IBMFS, but the review includes genetical changes, various types of signalling systems, while cytokines take only very little part in it.  

2.      Figure 1. is showing not generous defects underlying the IBMFS, but genetical ones.

In addition, the Fig. 2 and related information of this chapter would be better to move to the chapter near the the Fig. 1, which would show other mechanisms beside the genetical and would be more informative.

3.      Table 2 – the references are missing confirming these pro- and anti-effects.  

4.      Fig.2 is showing the five main types of IBMFS, while previous text is mentioning four main types of IBMFS. Fig. 2. description of figure should be in the text, not in the legend.

All together, the paragraph No. 7. Common Signaling Themes – is about everything and nothing and can be combined with the individual types of IBMFS or other chapters as previously suggested.

5.      Paragraphs 2-5 should be used as a subchapters of the main paragraphs indicating the five main types of IBMFS (as indicated in the Fig. 2). All other types of IBMFS can be in the separate paragraph. The conclusions are also mentioning only 4 main types of IBMFS.

6.      Each paragraph like FA, DC and other better to divide into a subchapters indicating the information, for example: genetical mutations, signalling pathways, inflammatory molecules or other, which makes easier to understand what the information is about. Very many abbreviations also hardens reading that should be explained as clear as possible. 

7.      Conclusions are superficially written and need more clear and scientifically important statements.

The English language is good, only some sentences need an additional checking.

Author Response

Reviewer 1

  1. The title does not matchthe whole review, i.e. it says that everything is about the role of inflammatory cytokines in the IBMFS, but the review includes genetical changes, various types of signalling systems, while cytokines take only very little part in it.  

Response: We have modified the tittle as follows: Revised title: Molecular and genetic mechanisms of inherited bone marrow failure syndromes: the role of inflammatory cytokines in the pathogenesis

  1. Figure 1. is showing not generous defects underlying the IBMFS, but genetical ones. 

Response: The caption of Figure 1 was revised in response to the comment as follows: Figure 1. Landscape of genetic alterations in IBMFS.

In addition, the Fig. 2 and related information of this chapter would be better to move to the chapter near the the Fig. 1, which would show other mechanisms beside the genetical and would be more informative. 

Response: Figure 2 was moved toward the Introduction and was cited in this section.

  1. Table 2 – the references are missing confirming these pro- and anti-effects.  

Response: References were added.

  1. Fig.2 is showing the five main types of IBMFS, while previous text is mentioning four main types of IBMFS. Fig. 2. description of figure should be in the text, not in the legend.  All together, the paragraph No. 7. Common Signaling Themes – is about everything and nothing and can be combined with the individual types of IBMFS or other chapters as previously suggested.

Response: We moved the paragraph to an earlier position so to orient the reader to common signaling biology.

  1. Paragraphs 2-5 should be used as a subchapters of the main paragraphs indicating the five main types of IBMFS (as indicated in the Fig. 2). All other types of IBMFS can be in the separate paragraph. The conclusions are also mentioning only 4 main types of IBMFS.

Response: We structured sections/subsections accordingly. We have discussed four common IBMFS throughout the manuscript. We removed SCN from Fig. 2 to avoid misunderstandings 

  1. Each paragraph like FA, DC and other better to divide into a subchapters indicating the information, for example: genetical mutations, signalling pathways, inflammatory molecules or other, which makes easier to understand what the information is about. Very many abbreviations also hardens reading that should be explained as clear as possible.  

Response: We structured subsubsections accordingly. We have limited the use of abbreviations (ROS, HR, ICL, DSB, and NHEJ were spelled out).

  1. Conclusions are superficially written and need more clear and scientifically important statements. 

Response: We have augmented the conclusions.

Reviewer 2 Report

This is a very interesting and well written review om a very timely topic. The figures are good. Inherited Bone Marrow Failure Syndromes (IBMFS) encompass various rare genetic disorders including Fanconi anemia, Diamond-Blackfan anemia, Shwachman-Diamond syndrome, dyskeratosis congenita, severe congenital neutropenia, and others like GATA2 deficiency and SAMD9/9L mutations. These syndromes are primarily characterized by distinct clinical features. Recent advances in exome sequencing have confirmed their categorization, revealing gene mutations that impact crucial cellular processes like DNA damage response (Fanconi anemia), ribosome structure (Diamond-Blackfan anemia), ribosome assembly (Shwachman-Diamond syndrome), and telomere maintenance (dyskeratosis congenita). The exact mechanisms behind IBMFS are not yet fully understood, but a prevailing hypothesis suggests that various stressors trigger TP53-dependent growth arrest and cell death in hematopoietic stem, progenitor, and precursor cells. This summary examines IBMFS, proposes a potential role for pro-inflammatory cytokines like TGF-β, IL-1β, and IFN-α in causing cytopenias, discusses their possible involvement in the development of myeloid neoplasia, and suggests the potential application of anti-inflammatory therapies.

Vasculature in bone plays important role in bone health and diseases including during the bone marrow failure. Moreover, vasculature is an important source of cytokines. PMID: 36669473; PMID: 33536212 Given the importance of the vasculature, authors should include a section on the role of bone vasculature and cite these and other references.

Minor checks required

Author Response

Reviewer 2

Vasculature in bone plays important role in bone health and diseases including during the bone marrow failure. Moreover, vasculature is an important source of cytokines. PMID: 36669473; PMID: 33536212 Given the importance of the vasculature, authors should include a section on the role of bone vasculature and cite these and other references.

Response: We have added this section.

Round 2

Reviewer 1 Report

The manuscript has been improved and can be published. The abbreviation of reactive oxygen species (ROS) is well known all over the world and it is overdone. The lines 68-77 explaining Fig. 2 is better to move before the Fig. 2, but this can be done by the journal's technical staff.